# Sentiment Analysis in Tamil and Tulu with Re-Ranking Enhanced BERT

Ratnavel Rajalakshmi[1], Bhuvana J[2], and Santhosh R[1]

[1] School of Computer Science and Engineering, Vellore Institute of Technology, Chennai
[2] Dept. of Computer Science and Engineering, Sri Sivasubramanian College of Engineering, Chennai
Corresponding author: rajalakshmi.r@vit.ac.in

**Abstract.** Sentiment analysis is essential in understanding the opinion of the people via their social media comments. This work focuses on sentiment analysis of Tamil and Tulu social media comments, by classifying the posts into positive, negative, mixed and neutral sentiment categories. The datasets are sourced from the DravidianLangTech@NAACL 2025 Shared Task on Sentiment Analysis in Dravidian Languages.
The proposed sentiment analysis framework uses fine-tuned multilingual transformer models and use Class Balanced Learning (CBL) to handle the class imbalance problem in the training set. A passage-re-ranking mechanism is also used in combination to align the CLS embedding with sentiment-specific features for decision making, thereby improving classification performance on noisy code-mixed text. The results show that class-balanced fine-tuning has considerably enhanced the sentiment prediction for low-resource, code-mixed text. This approach would serve as a strong foundation for future research in emotion detection and cross-lingual sentiment modeling within Dravidian languages.

**Keywords:** BERT· Re-ranking · CBL · Sentiment Analysis· Tamil · Tulu

## 1  Introduction

The rapid expansion of online social interaction among millions of individuals presents substantial hurdles for social media platforms. The essential analysis is crucial in the field of natural language processing studies, offering insights into the intricate dynamics of internet-based communication and interactions [3]. Linguistic inconsistency and variations due to transliterations are the primary causes of difference between the structure of monolingual Tamil and monolingual Tulu social media comments.

Sentiment analysis involves discerning emotions, feelings, or affection expressed in a given text, sentence, or paragraph [?]. The objective of the DravidianLangTech@NAACL 2025 [16] Shared Task is to determine the sentiment polarity of a social media posts and comments written in Tamil and Tulu. It is

noted that the constant increase in web and social media usage globally, leading to a significant rise in textual data expressing opinions. This publicly available data offers valuable insights that can be applicable in diverse domains, including marketing, finance, politics, and security. These insights present an excellent opportunity for individuals and businesses to understand user opinions better, enabling them to make informed decisions about improving their brands and services [4]. However, performing sentiment analysis on low-resource languages such as Tamil and Tulu is more challenging due to the limited availability of benchmark datasets and the ineffectiveness of available language-specific or multilingual or models [5]. Sentiment analysis can be conducted at various levels such as sentence, document, aspect, and phrase levels, describing human emotional states.

In this work, a class balanced learning approach is proposed by combining re-ranking enhanced BERT to address the challenges in for Sentiment analysis of Tamil and Tulu social media comments.

## 2    Related Work

Sentiment analysis has been a well explored method for English language and recently it has been extended to many low-resource languages, especially Dravidian Languages such as Tamil, Telugu, Tulu, Malayalam etc. Sentiment analysis is also studied for Hindi language [6] which involved a three-step fallback model. This model integrates machine translation, sentiment lexicons, and classification techniques to address the challenge of limited conventional text corpora in code-mixing computing models. Orthographic information plays an important role in cleaning code-mixed noisy content at the word level [7]. This approach involves removing noise and improving the quality of code-mixed text by considering the orthographic characteristics of individual words. The framework achieved optimal performance when utilizing unigram features. Kumar et al. employed deep learning models for text classification to detect sentiments in Manglish and Tanglish [9] [10]. An ensemble-based approach was used to analyse code-mixed tweets [11]. A classifier is developed which is capable of utilizing hand-engineered lexical, sentiment, and metadata features to differentiate between "positive," "neutral," and "negative" sentiments [12]. Cross-lingual embeddings are applied for analyzing code-mixed social media text, employing an unsupervised approach that resulted in an F1 score of 0.6355 [13]. Sentiment analysis on transliterated text using the RNN-LSTM technique was done [5] to retrieve sentiments from transliterated text. A computational approach was introduced to develop a Bengali version of Sentiment WordNet by using existing English sentiment lexicons along with a bilingual English–Bengali dictionary [14].

## 3    Methodology

The overall framework of the proposed sentiment analysis system for Tamil – English and Tulu–English code-mixed data is illustrated in Fig. 1. The work-

flow consists of four major stages: (i) data acquisition and pre-processing, (ii) feature extraction using multilingual transformer-based encoders, (iii) sentiment classification using fine-tuned BERT variants, and (iv) performance evaluation.

Initially, the raw comments collected from social media are preprocessed to remove noise from the comments, including converting comments to lowercase, removing special characters, replacing emojis with related words, and eliminating repeating characters that appear more than twice in a word. After pre-processing, the textual data are tokenized using a WordPiece tokenizer and transformed into dense vector embeddings by pretrained multilingual transformer models.

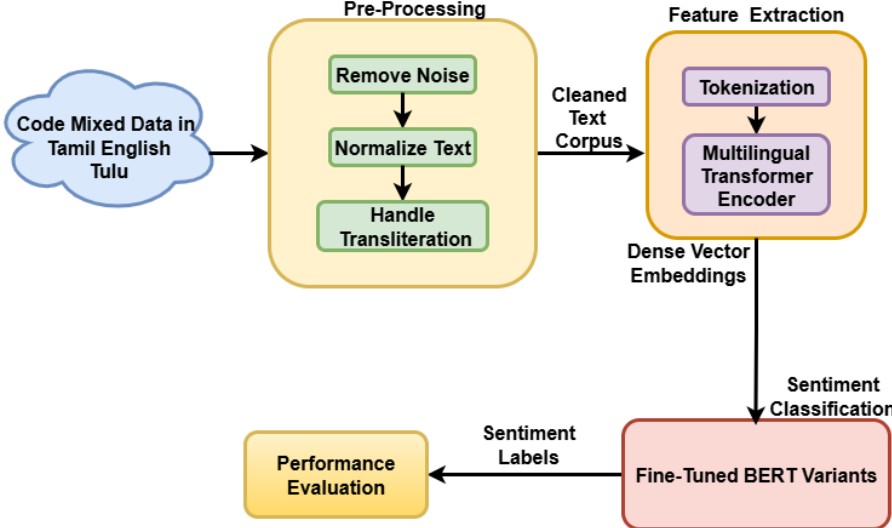

Fig. 1: Sentiment analysis framework for Tamil and Tulu comments.

*Model Description:* Model-1 utilizes a BERT multilingual-passage-re-ranking approach [?] for single-sentence sentiment classification in Tamil and Tulu.. The BERT model is fine-tuned using the training data, incorporating special tokens like CLS and SEP, and applying a word-piece tokenizer for tokenization. Token fusion involves summing up embeddings, positions, and segments, followed by the 12 encoders of the transformer to extract the hidden state vectors of the final stage. The CLS token's hidden vector is taken as the comment representation (V1), while feature vectors (V2) are generated for each comment based on sentiment word occurrences. Concatenating V1 and V2 (V = V1$\oplus$ V2), they are fed to a fully connected layer with softmax activation to predict the sentiment $\hat{y}$. This model fine-tunes the BERT architecture for single-text classification. The re-ranking-based adaptation is implemented by considering the [CLS] token embedding as a relevance score for each sentiment category, helping the pretrained

re-ranking architecture to serve as a single-sentence sentiment classifier. Class-Balanced Loss (CBL) is also incorporated to reduce the impact of minority categories while maintaining better performance across majority classes. The final output layer is a fully connected classifier with softmax activation.

Model-2 uses multi-lingual BERT classifier by incorporating Class-Balanced Loss (CBL) instead of the standard cross-entropy objective to handle severe class imbalance in the sentiment categories. CBL reassigns the weights to each sample based on its class frequency so that minority sentiment labels contribute proportionally to the optimization. The objective function is formulated as:

$$\mathcal{L}_{CBL} = -\alpha_t(1 - p_t)^\gamma \log(p_t) \tag{1}$$

where $p_t$ is the predicted probability for the true class, $\alpha_t$ is the class-balance weight determined by the inverse of the sample size, and $\gamma$ guides the focus on hard-to-learn samples.

## 4    Experiments

To evaluate the performance of the proposed approach, various experiments were conducted as detailed below.

*Dataset description* Two datasets Tamil [2] and Tulu [1] were used in this work. This sentiment analysis dataset contains different polarities viz., positive, negative, neutral and mixed feelings. Additionally, the dataset exhibits class imbalance, with the negative and mixed labels having fewer samples compared to the positive and negative class labels. Table 1 shows statistics of class-wise distribution for both Tamil and Tulu.

Table 1: Dataset statistics for Tamil and Tulu sentiment categories [16]

| Label | Train Set | | Validation Set | | Test Set | |
|---|---|---|---|---|---|---|
| | Tamil | Tulu | Tamil | Tulu | Tamil | Tulu |
| Positive | 18,145 | 3,769 | 2,272 | 470 | 1,983 | 453 |
| Negative | 4,151 | 843 | 480 | 118 | 458 | 88 |
| Neutral | 5,164 | 3,175 | 619 | 368 | 593 | 343 |
| Mixed | 3,662 | 1,114 | 472 | 143 | 425 | 120 |

For our experiments on both Tamil and Tulu, various batch sizes were attempted, specifically 16, 32, and 64. Adam optimizer with lr=3e-5 and eps=1e-8. A dropout rate of 0.4 was used to prevent overfitting, and Class-Balanced Loss was used to handle the sentiment class imbalance in Tamil and Tulu datasets. The final classification layer consists of 4 neurons, corresponding to the sentiment labels: Positive, Negative, Neutral, and Mixed.

BERT [?] underwent initial training on English language texts and later expanded to mBERT [?]. mBERT processes input tokens of up to 512, producing a 768-dimensional output vector with 12 attention heads. Our analysis includes only multi-lingual passage re-ranking model to assess improvements, especially for code-mixed texts. For this study, we utilized the BERT-base, Multilingual BERT(mBERT), and bert-multilingual-passage-re-ranking and mbert-base-uncased with CBL for Tamil and Tulu dataset, selected based on their performance on the DravidianLangTech@NAACL 2025 dataset.

## 5   Results and Analysis

The experimental evaluation compares multiple pretrained transformer models on both Tamil–English and Tulu–English datasets. Performance was measured using precision, recall, and F1-score. Table  2 summarizes the key metrics.

*Model-wise Comparison:* For Tamil, the Model-1: BERT Base Uncased + Re-ranking + CBL has given the best performance with an F1-score of 0.58 and has shown improvement over the standalone BERT + CBL (Model-2) which attained 0.50. This infers that the introduction of re-ranking improves the discriminative sentiment learning. Class-Balanced Learning applied has also shown consistent improvement for minority sentiment categories, particularly helping the Mixed and Negative sentiment categories For Tulu, Model-1 again has obtained the highest performance with 0.53 for F1, thereby outperforming Model-2 which has 0.41 for F1. The multilingual baseline mBERT has observed to have a competitive F1 score for Tulu as 0.52, but has dropped significantly in Tamil with 0.29 for F1. This may be due to the presence of increased spelling variations and transliteration noise. Our proposed approach has outperformed some of the baseline results [16] by approximately +0.03–0.05 F1, showing the effectiveness under low-resource and imbalanced settings.

*Language-wise Performance* From the results it can be observed that Tamil has achieved a slightly higher performance than Tulu with the proposed Model-1 achieving a 0.58 vs. 0.53 for F1. This improvement is achieved due to the reason that Tamil having comparatively richer pretrained model representations available in multilingual encoders. However, Tamil comments still possess reasonable transliteration noise and informal spellings, which contributed to the misclassifications in Neutral and Mixed sentiment categories. For Tulu, the lower F1-score may be due to its severe low-resource nature and the limited presence of pretrained linguistic knowledge. However, the competitive performance of mBERT on Tulu with 0.52 for F1 emphasizes the presence of cross-lingual representation across the Dravidian languages. Most errors are observed between the Neutral and Mixed sentiment categories, especially when comments have negations or emojis. Future improvements could use contextual sentiment lexicons or emoji-aware encodings to reduce this issue. Class-balanced optimization enhances the generalization of models, especially for smaller classes. This infers that handling

class imbalance problem is essential for sentiment modeling in low-resource scenarios.

Table 2: Performance comparison of baseline and CBL-enhanced BERT models for Tamil and Tulu sentiment classification.

| Model | Tamil | | | Tulu | | |
|---|---|---|---|---|---|---|
| | P | R | F1 | P | R | F1 |
| Model-1: BERT Base Uncased + Re-ranking+ CBL (Proposed) | **0.59** | **0.63** | **0.58** | **0.52** | **0.57** | **0.53** |
| Model-2: BERT Base + CBL | 0.56 | 0.58 | 0.50 | 0.54 | 0.45 | 0.41 |
| mBERT Multilingual Uncased | 0.44 | 0.29 | 0.29 | 0.52 | 0.56 | 0.52 |

## 6    Conclusion

This study presented a comprehensive framework for sentiment analysis of Tamil and Tulu social media data using multilingual transformer-based architectures. BERT multilingual passage re-ranking model with the combination of Class Balanced Loss has enhanced the performance of the proposed approach. Experimental results observed that using CBL significantly enhanced the model's ability to address class imbalance, yielding an F1-score of 0.58 for Tamil and 0.53 for Tulu on the test sets. This infer that noisy Tamil data is still difficult to process for sentiment classification and the ability of the multi-lingual transfer learning for Tulu. Future work will explore extending the current framework to perform fine-grained emotion detection in Tamil, Tulu, and other Dravidian languages.

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
