# OpenReview forum: "Sentiment Analysis in Tamil and Tulu with Re-Ranking Enhanced BERT"
_SPELLL.org/2025/Workshop/LC4 — LC4 2025 Oral_

### Official Review · Reviewer_vF2y · 2025-11-03
**Interesting and relevant direction for low-resource sentiment processing, but the current performance and analysis do not yet meet the bar for strong acceptance**

**Rating:** 5
**Confidence:** 4

**Review:**

This paper tackles sentiment analysis for code-mixed Tamil–English and Tulu–English text using multilingual transformer models, including a Class-Balanced Learning (CBL) extension. The topic is important and timely, especially given the limited availability of computational resources for Dravidian languages. The motivation is clear, and the overall workflow is explained with appropriate figures and tables.

The strengths are the focus on low-resource linguistic settings, a comprehensive comparison across multiple multilingual pretrained models, and the observation that CBL improves handling of class imbalance. The paper also demonstrates practical challenges such as transliteration and noisy expressions in social media text.

However, the work has several weaknesses that limit its impact:

Technical and Results Concerns

Test performance for Tamil is very low (F1 ≈ 0.21), calling into question the usefulness of the proposed solution in real deployment.

Lack of strong baselines from prior Dravidian shared tasks makes it difficult to assess improvement over published work.

Key architecture details are insufficiently described particularly the passage-reranking model and its relevance to single-sentence sentiment classification.

Evaluation inconsistencies exist (training vs. testing divergence not analyzed).

Clarity and Reproducibility Issues

Writing quality needs revision: grammatical errors, repetition, and missing contextual explanations occur throughout.

Dataset details (source, licensing, preprocessing rules) should be more clearly documented.

No ablation studies or statistical significance checks to justify architectural choices.

Suggested Improvements

Include prior top-performing FIRE/DravidianCodeMix baselines.

Provide deeper error analysis for confusion between mixed/unknown sentiments.

Strengthen methodology: explicit handling of transliteration, lexicon features, and contextual signals.

Improve writing structure and remove redundancy.

---

### Official Review · Reviewer_nxmw · 2025-11-03
**Sentiment Analysis in Tamil and Tulu using Transformers**

**Rating:** 3
**Confidence:** 2

**Review:**

Authors proposed transformer based reranking method on Sentiment analysis on Dravidian languages.
1) The dataset cited in abstract shows the sarcasm dataset from Dravidian CodeMix Fire 2024, but the paper deals with sentiment analysis.
2) There is a ambiguity in category names of the dataset, somewhere it is mentioned as Neutral state, but in statistics it has not mentioned.
3) How reranking method is applied and after balance what is the count of the dataset?
4) Test set description is not mentioned in the paper.
5) Test set results projected here is 0.21, then what is the use of the proposed architecture?
6) For tables citation and discussion is missing.